# Chelation Therapy Associated with Antioxidant Supplementation Can Decrease Oxidative Stress and Inflammation in Multiple Sclerosis: Preliminary Results

**DOI:** 10.3390/antiox12071338

**Published:** 2023-06-24

**Authors:** Alessandra Vezzoli, Simona Mrakic-Sposta, Cinzia Dellanoce, Michela Montorsi, Daniele Vietti, Maria Elena Ferrero

**Affiliations:** 1Institute of Clinical Physiology, National Research Council (IFC-CNR), Piazza Ospedale Maggiore 3, 20159 Milano, Italy; cinziacarla.dellanoce@cnr.it; 2Department of Human Sciences and Promotion of the Quality of Life, San Raffaele Roma Open University, Via di val Cannuta 247, 00166 Roma, Italy; michela.montorsi@uniroma5.it; 3Driatec Srl, Via Leonardo da Vinci 21/E, 20060 Cassina de’ Pecchi, Italy; danielevietti@gmail.com; 4Department of Biomedical Sciences for Health, Università degli Studi di Milano, Via Mangiagalli 31, 20133 Milano, Italy; mariaelena.ferrero@unimi.it

**Keywords:** EDTA, neurotoxicity, thiols’ redox status, oxidative damage, cytokines, ROS, EPR, neurodegeneration

## Abstract

An imbalance of oxy-inflammation status has been involved in axonal damage and demyelination in multiple sclerosis (MS). The aim of this study was to investigate the efficacy of an antioxidant treatment (calcium disodium ethylenediaminetetracetic acid—EDTA) chelation therapy associated with a micronutrient complex in MS patients. A total of 20 MS patients and 20 healthy subjects, enrolled as a control group (CTR), were recruited. We measured the plasma ROS production and total antioxidant capacity (TAC) by a direct assessment using Electron Paramagnetic Resonance; activities of the antioxidant system (thiols’ redox status and enzymes); and the urinary presence of biomarkers of oxidative stress by immunoenzymatic assays. We also evaluated the levels of inflammation by plasmatic cytokines (TNFα, IL-1β, and IL-6) and assessed the sICAM levels, as well as the nitric oxide (NO) catabolism and transthyretin (TTR) concentration. Comparing CTR and MS, in the latter ROS production, oxidative damage, inflammatory biomarkers, and NO metabolite concentrations results were significantly higher, while TAC was significantly lower. Treatment in MS induced significant (*p* < 0.05) down-regulating of pro-inflammatory sICAM1, TNF-α, IL6, as well as biomarkers of lipid peroxidation and DNA damage production. The protective effect exhibited may occur by decreasing ROS production and increasing antioxidant capacity, turning into a more reduced thiols’ status.

## 1. Introduction

Multiple sclerosis (MS) is a chronic immune disease of the central nervous system (CNS), characterized by demyelination [1], where inflammation is a major driver of the pathology. Besides, studies have demonstrated that MS patients have an accumulated deposition of iron/metals in CNS, particularly in gray and white matter structures [2,3]. However, the actual cause of the neurodegenerative components of the disease is unclear [4,5].

Recent observations confirm the fundamental role of oxidative stress (OxS) that is considerably involved in a lot of pathological features of MS such as myelin and axonal degeneration and inflammation [6,7]. In addition, OxS promotes an existing inflammatory response and contributes to tissue injury. Excessive or sustained Reactive Oxygen Species (ROS) levels are involved in the pathogenesis of MS [6,8]. ROS are chemically high-reactive molecules formed as a natural byproduct of oxygen metabolism, and they are significantly implicated in homeostasis and cell signaling. However, ROS levels can increase dramatically in pathological conditions, leading to significant damage to cell structures. Due to the high reactivity and short life of ROS, they are difficult to detect directly in biological tissues [9,10]. Several oxidized molecules can potentially be used as diagnostic biomarkers [11] assessed in biological fluids. Indeed, ROS can affect important classes of biological molecules, thus leading to multiple lipids, proteins, and DNA via peroxidation and/or nitration processes [12]. 

Disturbance of reduction–oxidation (redox system) metabolism has a key importance in MS pathogenesis [13]. The maintenance of the redox homeostasis is effectively supported by a complex system of antioxidants. This includes both enzymatic and nonenzymatic compounds. The antioxidant system capacity can be adapted by increasing the levels of enzymes such as superoxide dismutase (SOD), catalase (CAT), and glutathione peroxidase (GSHPx). A positive correlation between SOD levels and the Expanded Disability Status Scale (EDSS) score has been reported [14], indicating the importance of antioxidant defense to control MS disability. Interestingly, persistent hyperactivation of oxidative enzymes suggests an “OxS memory” in chronic neuroinflammation [15].

Glutathione (GSH) represents one of the most important and ubiquitous among the nonenzymatic antioxidants. Indeed, GSH and related thiols (cysteine (Cys) and cysteinylglycine (CysGly)) play a fundamental role in the antioxidant defense system, interacting via redox and disulfide reactions, and become part of a dynamic system referred to as the redox thiols’ status [16]. Undoubtedly, ROS signaling is a major contributor in the organism’s defense system, but if homeostasis is breached, a vicious circle, that comprises inflammation and degeneration, initiates. 

It has already been reported that IL-6, TNFα, and IL-1β, are implicated as mediators of multiple sclerosis pathology. Axon injury may be triggered by the biological actions of increased cytokines; in particular, disease progression has been associated to altered levels of TNFα and IL-6 in cerebrospinal and/or serum [17,18,19]. Intercellular adhesion molecule-1 (sICAM-1) is a cell surface glycoprotein expressed at a low basal level in immune, endothelial, and epithelial cells, but in response to inflammatory stimulation, it is up-regulated [20]. Therefore, sICAM levels might be adopted as a monitoring marker of therapy or inflammatory response [21]. 

The induction of the activation of microglia and mitochondrial dysfunction plays a particular role in inflammatory processes. In fact, studies present in the literature have reported that the demyelination and axonal damage can be initiated by the activation of microglia, which releases Nitric Oxide (NO) [22]. Moreover, inducible nitric oxide synthase (iNOS), an enzyme with cytotoxic effect, can be induced by TNFα, IL-1β, and INFγ. iNOS is present in actively demyelinating lesions, and the level of stable reaction products of nitric oxide, i.e., NO metabolites (NOx = nitrite + nitrate), are significantly higher in the plasma or serum of patients with MS [23,24]. 

Significant expression differences were detected in serum proteins, and transthyretin (TTR) was identified as potential disease signatures for MS patients [25]. The liver’s parenchymal cells synthesized TTR and secreted it into the plasma, whereas in the CSF, it originates mainly from the choroid plexus in the ventricles [26]. TTR is one of the main transporters of the retinal binding protein that is a transporter of vitamin A [27]. After peripheral nerve injuries, TTR is able to enhance neurite outgrowth and nerve regeneration [28], resulting as beneficial to the Peripheric Neuron System.

The aim of therapeutic treatments for MS should be the restoration of general homeostasis, including redox balance, to prevent the reverting of physiological ROS signaling. Indeed, many of the approved MS therapeutics lead to decreased Oxs, and this effect seems to contribute to their efficacy. However, different strategies that interfere with OxS failed in clinical evaluation. Nevertheless, antioxidants may show to be beneficial as co-treatments with anti-inflammatory reagents, resulting in a superior clinical outcome. Calcium disodium ethylenediaminetetraacetic acid (CaNa_2_EDTA) (hereafter, EDTA) chelation therapy was proposed as an option in the treatment of neurodegenerative disorders and/or diseases associated with metal burdens [29,30]. It was shown that chelation therapy with EDTA has many important functions, including antioxidant and anti-inflammatory properties, as well as the capacity to protect against endothelial damage [3,30]. To note that, chelation therapy has been demonstrated as well-tolerated and effective [31]. Moreover, oral GSH supplements during chelation therapy have been reported to counteract the depletion of endogenous GSH in patients affected by neurodegenerative diseases [32].

The aim of this study was to evaluate the efficacy of an antioxidant treatment on: (a) ROS production directly assessed by Electron Paramagnetic Resonance (EPR); (b) the activities of the oxidative and antioxidative system and the presence of degradation and end products derived from cellular components, i.e., biomarkers of oxidative stress; (c) the biomarkers of systemic inflammatory response in plasma (i.e., levels of TNFα, IL-1β, IL-6, sICAM); (d) nitric oxide catabolism; and (e) a marker of neurologic disease (i.e., transthyretin—TTR).

## 2. Materials and Methods

### 2.1. Subjects

In this study, which was carried out to evaluate the effects of a specific antioxidant treatment, 20 MS patients (10F/10M, age 39.2 ± 10.5, weight 63.6 ± 11.9, height 1.68 ± 0.07) and 20 age- and gender-matched healthy controls (Ctr: 10F/10M, age 33.9 ± 11.1, weight 64.4 ± 12.3, height 1.68 ± 0.09) were recruited. 

MS patients had clinically defined MS based on magnetic resonance imaging (RMI). All spontaneously chose to start chelation therapy. Seven of them had interrupted previous therapies with conventional drugs against MS almost 3 months before starting chelating treatment. The other MS patients had never been previously treated with drugs. 

In healthy subjects, exclusion criteria concerned: alcohol, obesity, smoke, current use of medicines, special diet, minerals, vitamins, or other kind of supplementation, as well as antioxidant supply.

All subjects provided written informed consent. All procedures followed the Declaration of Helsinki guidelines and were approved by Milan University’s Ethics Advisory Committee (number 64/14). Personal data were recorded and analyzed in an anonymous format. All MS patients selected and enrolled for the study received chelation therapy once a week.

The subjects assigned to the control group did not undergo any intervention in order to evaluated basal differences between the healthy vs. disease groups at pre-treatment.

### 2.2. Study Design

To verify the possible toxic-metal burden of the patients, they were submitted to a “chelation test” (see below), before starting the chelation therapy, that lasted three months.

Another chelation test was conducted on the subjects after ten applications to assess body burden modifications; indeed, they were monitored for toxic-metal burden throughout chelation therapy. Patients initiated weekly chelation therapy after the first “chelation test” showed a toxic-metal burden.

### 2.3. Chelation Test

A 500 mL physiological saline solution (Farmax srl, Brescia, Italy), containing CaNa2EDTA (2 g), was slowly (about 2 h) administered intravenously to MS patients. Urine sample collection was carried out before, after, and in the 12 h following chelation treatment. The Laboratory of Toxicology (Doctor’s Data Inc., St. Charles, IL, USA) analyzed the urine samples, carefully collected in sterile vials, as previously reported [29].

An acid-digestion of the samples was conducted with certified metal-free acids in a closed-vessel microwave digestion system. Samples were then diluted with ultrapure water and examined via inductively coupled plasma mass spectrometry (ICP-MS), a reliable method to reduce interference that uses collision/reaction cell methods coupled with ion-molecule chemistry. Urine standards, both certified and in-house, were used for quality control and data validation. Results were reported in micrograms per g of creatinine to prevent the possibly great margin of error due to fluid intake and sample volume.

Twenty toxic metals were analyzed: tungsten (W), uranium (U), thorium (Th), thallium (Tl), tellurium (Te), tin (Sn), antimony (Sb), platinum (Pt), palladium (Pd), lead (Pb), nickel (Ni), mercury (Hg), cesium (Cs), cadmium (Cd), bismuth (Bi), beryllium (Be), barium (Ba), arsenic (As), aluminum (Al), and gadolinium (Gd); the latter was commonly used as a contrast agent in magnetic resonance imaging to diagnose MS.

### 2.4. Antioxidant Treatments

Commercial multivitamin complex and redox glutathione (Oximix MV+ and Oximix 7+ detox, respectively, DRIATEC, Milan, Italy) were assumed every alternate day by MS patients at the beginning of chelation treatment as a micronutrient complex.

Oximix 7+ detox showed good results, increasing the significantly reduced GSH level in neurodegenerative disease subjects in a previous paper [32]. Oximix MV+ is a complete multivitamin and multi-mineral supplement together with a lower amount of glutathione, but with the other antioxidants needed for a complete reduction of the pool. The use of a single antioxidant element could result in pro-oxidant or counterproductive over 60 days of use [33]. In detail, vitamin E and beta carotene work as antioxidants on the cellular membrane against the propagation of the damage of free radicals. These vitamins can be reduced and reutilized by vitamin C and glutathione, which can be reduced again with the help of lipoid acid and Q10 [34]. As activators of the enzymes, we need selenium for the glutathione peroxidase and vitamin B2 for the GSH reduction, while manganese, copper, and zinc are used for SOD and catalase.

The subjects assigned to the control group did not undergo any intervention.

### 2.5. Blood, and Urine Samples

The subjects assigned to the control group underwent a basal evaluation, while MS patients underwent two test sessions at study entry and three months after enrollment (T0, T1) for blood and urine sample collection (Figure 1). For each subject, in the morning before breakfast, venous blood samples (about 5 mL) were drowned in EDTA and LH tubes (Vacuette tube, Greiner bio-one, Kremsmünster, Austria). Blood samples were centrifugated (Hettich^®^ MIKRO 200R centrifuge) for 10 min to separate plasma and red blood cells (RBC). Multiple aliquots were immediately frozen and stored at −80 °C. Plasma samples were collected to determine levels of ROS, TAC, SOD, catalase, IL-6, TNFα, IL-1β, sICAM-1, and TTR. Aminothiols’ redox status was evaluated by an RBC analysis.

Urine samples were collected in a sterile container provided to the subjects by voluntary voiding. Nitrite/nitrate (NOx), lipid peroxidation (8-isoprostane), and DNA damage (8-OH-2-deoxyguanosine) concentrations were measured for urine samples.

All samples were stored in multiple aliquots at −80 °C until assayed. Samples were thawed only once before analysis, performed within two weeks from collection.

#### 2.5.1. ROS and Antioxidant Capacity by Electron Paramagnetic Resonance

ROS production and TAC were quantified by Electron Paramagnetic Resonance Spectroscopy in X-band at 9.3 GHz (EPR) (E-Scan Bruker, Billerica, MA, USA), as previously indicated [33,35,36,37,38]. The temperature of the analysis was 37 °C, controlled by the Temperature and Gas Controller ‘‘Bio III’’ (Noxigen Science Transfer & Diagnostics GmbH, Elzach, Germany), interfaced with the E-Scan [33,35,36,37,38]. Briefly, ROS were determined using CMH (1-hydroxy-3-methoxycarbonyl-2,2,5,5-tetramethylpyrrolidine) as a spin trap probe and stable radical CP·(3-Carboxy2,2,5,5-tetramethyl-1-pyrrolidinyloxy) as an external reference to convert ROS determinations into absolute quantitative values (μmol·min^−1^). 

1,1-diphenyl-2-picrylhydrazyl (DPPH•) quenching [39,40] was used to measure TAC. The calculated antioxidant capacity was expressed in terms of Trolox equivalent antioxidant capacity (TAC, mM).

#### 2.5.2. Thiols

Total (tot), reduced (red), and oxidized (ox) aminothiols (Cys: cysteine; CysGly: cysteinyl glycine; and GSH: glutathione) were measured in red blood cells, and homocysteine (Hcy) in plasma, according to previously validated methods [41,42,43]. 

Briefly, an isocratic HPLC analysis on a Discovery C-18 column (250 × 4.6 mm I.D, Supelco, Sigma-Aldrich, St. Louis, MO, USA) was used to perform thiol separation at room temperature, eluted with a solution of 0.1 M acetate buffer, pH 4.0: methanol, 81:19 (*v*/*v*), at a flow rate of 1 mL·min^−1^. A fluorescence spectrophotometer (Jasco, Japan) was used to measure fluorescence intensities, with an excitation wavelength at 390 nm and an emission wavelength at 510 nm. A standard calibration curve was used.

#### 2.5.3. Antioxidant Enzymes 

Superoxide dismutase (SOD) and catalase (CAT) were assessed by commercial Cayman Chemical kits (Ann Arbor, MI, USA, Cat. 70600 and Cat. 707002, respectively). The methods have been previously described [44].

#### 2.5.4. Inflammatory Markers

Interleukin-IL-6, IL-1β, and tumor necrosis factor-a (TNFα) levels were determined using ELISA assay kits (Cat. 501030, 583311 and 589201, respectively, Cayman Chemical, Ann Arbor, MI, USA), based on the double-antibody “sandwich” technique in accordance with the manufacturer’s instruction. The methods have been previously described in detail [36,37,43]. Plasma levels of inflammatory markers in pg/mL were then calculated according to the optical density of each well. 

#### 2.5.5. Intercellular Adhesion Molecule-1 sICAM-1

The human sICAM-1 ELISA assay kit (Cat. RAF102R, BioVendor, Brno, Czech Republic), an enzyme-linked immunosorbent assay, was utilized for the quantitative detection of sICAM-1 in plasma samples. The amount of sICAM-1 present was measured by absorbance at 450 nm. Concentrations were determined by a standard curve prepared from sICAM-1 dilutions. 

#### 2.5.6. Transthyretin (TTR)

Plasmatic transthyretin concentrations were assessed by a commercial kit (Cat. EH1025, Fine test, Wuhan, China) based on sandwich enzyme-linked immune-sorbent assay technology. The standards and test samples were read at 450 nm, and then the concentration of the target was calculated.

#### 2.5.7. Nitric Oxide Metabolites

Nitrite (NO_2_) plus nitrate (NO_3_) levels (NO_x_) were assessed in urine via a colorimetric method based on the Griess reaction [39,45], using a commercial kit (Cat. 780001, Cayman, Bertin Pharma, Montigny le Bretonneux, France).

#### 2.5.8. Oxidative Damage Biomarkers

Lipid peroxidation (8-Isoprostane—8-iso-PGF2α) and DNA damage (8-OH-2-deoxyguanosine—8-OH-dG) were determined in urine samples by commercial immunoassays (Cat. 515351 and 589320, respectively, Cayman Chemical, Ann Arbor, MI, USA). The methods have been previously described [36,37,41].

### 2.6. Statistical Analysis 

Data are presented as the mean ± standard deviation (SD). Statistical analysis was performed using the GraphPad Prism package for Mac (GraphPad Prism 9.5.0, GraphPad So ware Inc., San Diego, CA, USA) and IBM® SPSS Statistics software (IBM corporation, Armonk, NY, USA). After a normal distribution test, statistical analyses were performed using non-parametric tests. An ANOVA was performed, as well as Dunn’s multiple comparison tests to further check the among-groups significance. The effect of the chelation therapy and multivitamin complex/redox glutathione in SM subjects was tested using the Wilcoxon matched-pairs signed rank test. A *p* < 0.05 was considered statistically significant. dCohen with 95% CI was used for calculating the size effect in SM subjects. Change Δ% estimation (((pre-value − post-value)/pre value) × 100) is also reported in the text. ROS production was considered as the primary outcome (no other parameters were taken into account), and prospective calculations of power to determine the sample size were made using G power software (GPower 3.1) [46]. At 80% power, the sample size—calculated in preliminary studies [33,42]—was set at eleven/ thirteen subjects.

## 3. Results

Two MS subjects discontinued their study participation. The chelation tests showed that all MS patients had high toxic metals levels, and, as previously reported, these levels significantly decreased when measured at the end of treatment [47]. 

The levels of blood/urine markers examined in the present study were compared between MS patients and healthy controls and in MS patients before and after the antioxidant treatment.

### 3.1. ROS Production, Redox Status, and Oxidative Stress Biomarkers

The evaluation of ROS production (Figure 2A) and TAC (Figure 2B) plasma levels in the MS patients and control group revealed that the ROS in patients were higher than controls (+44%), while TAC in patients was lower than controls (−42%). Furthermore, in MS patients, after treatment, ROS production significantly decreased (−15%, dCohen = 0.657) and TAC increased (+30%; dCohen = 1.036). The levels of lipids peroxidation (8-iso PGF2α pg·mg^−1^ creatinine) and DNA oxidation (8-OH-dG ng·mg^−1^ creatinine) were significantly higher in patients than in the control group (Figure 2C +114% and Figure 2D +65%, respectively). After treatment, in MS patients, the levels of both 8-iso PGF2α (−41%; dCohen = 1.140) and 8-OH-dG (−32%: dCohen = 1.163) significantly decreased.

SOD and CAT activity are reported in Figure 2E,F, respectively. In the plasma of MS patients, SOD activity was significantly (+63%) higher than in the control group. Contrarily, no significant differences between the MS and control groups were recorded in plasma CAT activity. After treatment, the levels of both CAT (+25%; dCohen = 0.958) and SOD (+25%; dCohen = 0.526) significantly increased. 

In Figure 3, values of oxidized (ox) (Figure 3A–C), reduced (red) (Figure 3D–F), and total (tot) (Figure 3G–J) aminothiol concentration recorded in the control group and in MS patients at baseline (T0) and after 3 months (T1) of treatment are reported. Significant differences in aminothiol concentrations between MS patients at pre vs. control group were observed: lower red CysGly (E; −38%), reduced GSH (F; −71%), and GSH total (I; −13%) levels; and higher oxidized CysGly (B; +290%), oxi GSH (C; +40%), total CysGly (H; +130%), and total Hcy (J; +185%) levels in MS patients before treatment vs. the control group were reported. 

After treatment, the concentration of reduced oxi CysGly (B; −60%; dCohen = 3.235;), oxidized GSH (C; −9%; dCohen = 0.443), red Cys (D; +78%; dCohen = 1.455), red CysGly (E; +32%; dCohen = 0.705), red GSH (F; +103%; dCohen = 2.442), total CysGly (H; −49%; dCohen = 3.054), and total GSH (I; +6%; dCohen = 0.834) significantly changed. 

### 3.2. Inflammation Biomarkers

The assessment of the plasmatic IL-6 concentration of MS patients represented a higher level prior to antioxidant treatment compared to the healthy group (+238%), and the statistical analysis of the data were significant (Figure 4A). After 3 months of treatment, the IL-6 concentration of the patients significantly declined compared to pre-treatment values (−34%: dCohen = 0.812). Additionally, the assessment of the TNF-α (Figure 4B) and IL-1β (Figure 4C) concentrations in the plasma of MS patients represented higher levels in the MS patients (+626% and +152%, respectively) prior to antioxidant treatment compared to the healthy group. After 3 months, only TNF-α concentrations declined compared to before the treatment (−11%, dCohen = 0.556). 

sICAM concentration (Figure 4D) in the plasma of MS patients was significantly (+71%) higher in the MS patients prior to antioxidant treatment compared to the healthy group, but it decreased significantly (−24%: dCohen = 1.102) after treatment.

In Figure 5A, urinary nitric oxide catabolite concentrations are reported. The NOx level was statistically higher (+53%) in the MS group at baseline vs. the control group. Anyway, NOx showed a significant decrease after 3 months of treatment compared to the pre-treatment evaluation (−27%; dCohen = 1.087). The values of transthyretin concentration had a higher result, but not significant, (+12%) in MS patients with respect to the control group, and did not change after treatment (Figure 5B).

## 4. Discussion

The progression of MS is reported as directly related to oxidative stress level [48]. Indeed, demyelination and axonal damage, observed in MS, have been associated with an imbalance in redox status. It has been demonstrated that lipidic hydroperoxides, NO metabolite levels, and TAC in plasma may potentially be considered predictive parameters of disability and disease progression in MS [6,49]. 

The results of the present study not only confirm previously reported data, observing higher levels of biomarkers of lipid peroxidation, DNA damage, as well as lower levels of antioxidant capacity in patients with MS compared to healthy subjects, but, to the best of our knowledge, for the first time, we associate them with a higher ROS production directly assessed by EPR.

Moreover, the antioxidant treatment had an efficient result in MS patients, decreasing oxidative damage biomarker (8-OHdG and 8-isoprostane) levels, related to a lower release of ROS, and these phenomena are linked to an increase in total antioxidant capacity. 

The observed decrease of TAC values in the plasma of MS patients may be associated with variations in low-weight antioxidant molecules, and, in particular, in SH groups. Excessive changes in redox status, deriving from affected respiratory chain complex activity, may be the cause of the low SH amount reported in cerebrospinal fluid and plasma of MS patients [50]. Anyway, the role and amount of GSH in MS is almost not determined. Decrease, increase, and not significant differences in the GSH level in MS patients were previously reported [51]. Our results showed increases in the oxidized and decreases in reduced thiols’ forms in MS patients compared to healthy subjects. Data recorded would support the hypothetical antioxidant system global deficiency that was partially compensated by antioxidant treatment that may induce a turn to more reduced forms. 

In agreement with previous studies, we found that plasma homocysteine levels were increased in MS patients compared to healthy subjects [52,53,54,55]. High homocysteine levels are reported to be a risk factor for neurodegenerative diseases [56], neurological decline, and cognitive impairment [57,58,59] in MS. Thereafter, the reduction in Hcy recorded after treatment, even if not statistically significant, may have a beneficial effect. 

OxS activates a response as an adaptive mechanism in order to protect cells against ROS-induced toxicity and to maintain tissue redox balance. This stress response includes transcriptional activation of various genes, encoding antioxidant and detoxification enzymes. In MS patients’ SOD activity, because its tendency is to prevent detrimental effects and CNS injuries, SOD is upregulated by an altered redox homeostasis [60]. SOD level variations might be regulated by exceeding the ROS production that can have a direct influence on the SOD gene [61]. SOD is the most important antioxidant enzyme and plays the first defense against OxS [62]. Like many authors, [23,63] we found an increase of this most potent antioxidant enzyme activity. This is a consequence of the increased oxidation and an attempt to dampen its deleterious effects in MS. Upregulation of antioxidant activity represents one of the possible therapeutic approaches, and, here, the resulting adopted treatment available to this aim.

ROS can activate several factors of transcription, e.g., nuclear transcription factor-kappa B (NF-kB), which upregulates the expression of many genes involved in MS, such as TNF- α, nitric oxide synthase (iNOS), and sICAM-1 [64]. As previously reported, our data recorded higher levels of TNF-α in MS patients than in healthy subjects [65,66]. TNF-α plays a key role in the inflammation propagation, as sTNFR1 induces the activation and recruitment of immune cells. The latter can directly generate OxS-activating ROS and RNS production. Anyway, since sTNFR2 exerts a neuroprotective function and promotes tissue regeneration [67,68], the TNF-α unchanged levels that we observed in the MS patients after antioxidant treatment may be a way to support the TNF-α immunomodulation. 

The data reported here showed that IL-6, in line with previous studies, was elevated in the plasma of MS patients [69,70]. Our results showed significant changes in the plasma levels of TNF-α, IL-6, and sICAM 1, but not in IL-1β after 3 months of treatment. Similar results were obtained through the means of other kinds of therapeutical treatments [70,71,72]. The TNF-α and IL-6-persistent levels that were more elevated than in healthy subjects are probably due to the fact that these cytokines, usually considered proinflammatory mediators in MS pathology, contribute to neuroprotection too. Therefore, a reduction of these cytokines may be unfavorable in clinical MS treatments. Moreover, proinflammatory cytokines such as interleukin-1 (IL-1), IL-6, and TNF-α are potent SOD activators. 

Furthermore, we studied plasma levels of sICAM-1, and, as reported in previous works [73], it had been found significantly increased in patients with MS compared to healthy controls. sICAM-1 expression is upregulated by proinflammatory cytokines such as interleukin-1β, TNF-α. Increased sICAM content was related to inflammation, and often, sICAM-1 is used as a surrogate marker to monitor the response to therapy. Indeed, an overexpression of adhesion molecules, such as ICAM-1 on the endothelial surface, stimulates the adhesion of leukocytes, promoting inflammatory processes. Serum sICAM-1 has been postulated as a marker of MS relapse [74]. Changes in sICAM-1 in patients with relapsing and relapsing–progressive disease are reported related to the occurrence of attacks and increases in sICAM-1 concentration, preceded by the appearance of new gadolinium enhancing lesions [75]. Thereafter, the observed reduction in sICAM-1 levels after treatment may be considered a favorable effect. TNFα and IL-1β can induce the production of iNOS, an enzyme with cytotoxic effects, present in actively demyelinating lesions, and NO metabolites are reported as elevated in the CSF and serum of MS patients [76,77]. Moreover, a role for oxidizing molecules of NO in the immune-pathogenesis of MS has been suggested. In the present study, significantly higher levels of NO metabolites were registered in MS patients versus healthy subjects, as previously reported [65]. We found that antioxidant treatment decreased the NOx level, and this reduction in MS patients could be beneficial too [78].

Cys10 residue of TTR, due to its sensitivity to redox conditions, has been considered as a sensor of the environmental redox state [79]. In cerebrospinal fluid, the specific oxidative modification of TTR Cys10 has been associated with demyelinating diseases [80]. The serum TTR content was the same as in healthy persons, as previously reported [81]. Therefore, the not significant difference recorded in this study might suggest an unaltered level of TTR in MS patients’ cerebrospinal fluid too.

### Limitations

This study suffers from some limitations. The low number and the high inter-individual variability of examined subjects are certainly limits; moreover, the severity on each relapse and the time between each one was not evaluated. Contrarily, the strength of the work comes from the assessment of most oxy-inflammation biomarkers, and, particularly, for the first time, of ROS production and antioxidant capacity by EPR, as well as aminothiol redox status after treatment of chelation therapy, associated with antioxidant supplementation. Further investigations are needed to confirm our preliminary results. 

## 5. Conclusions

By preventing or reducing oxidative damage, we may potentially prevent or delay neurodegeneration as a core substrate of disability. The present finding of the adopted treatment causing downregulation of inflammatory expression and OxS may be considered effective. Its protective effect may be attributable to a decrease of ROS production and an increase in antioxidant activity. Considering the results obtained, it is possible to plan an MRI of the brain and/or spinal cord.

## Figures and Tables

**Figure 1 antioxidants-12-01338-f001:**
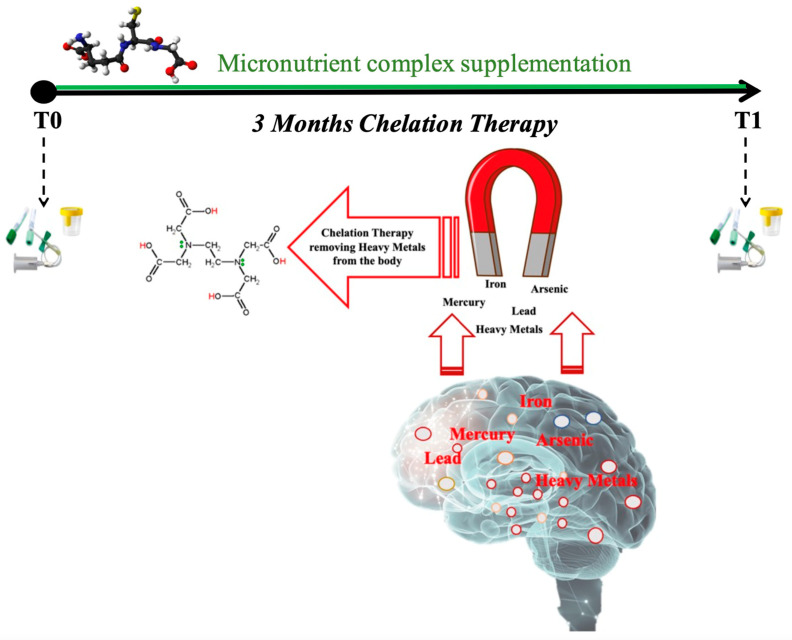
Schema of experimental protocol used to collect blood and urine samples before (T0) and after (T1) three months of chelation therapy in association with micronutrient complex supplementation.

**Figure 2 antioxidants-12-01338-f002:**
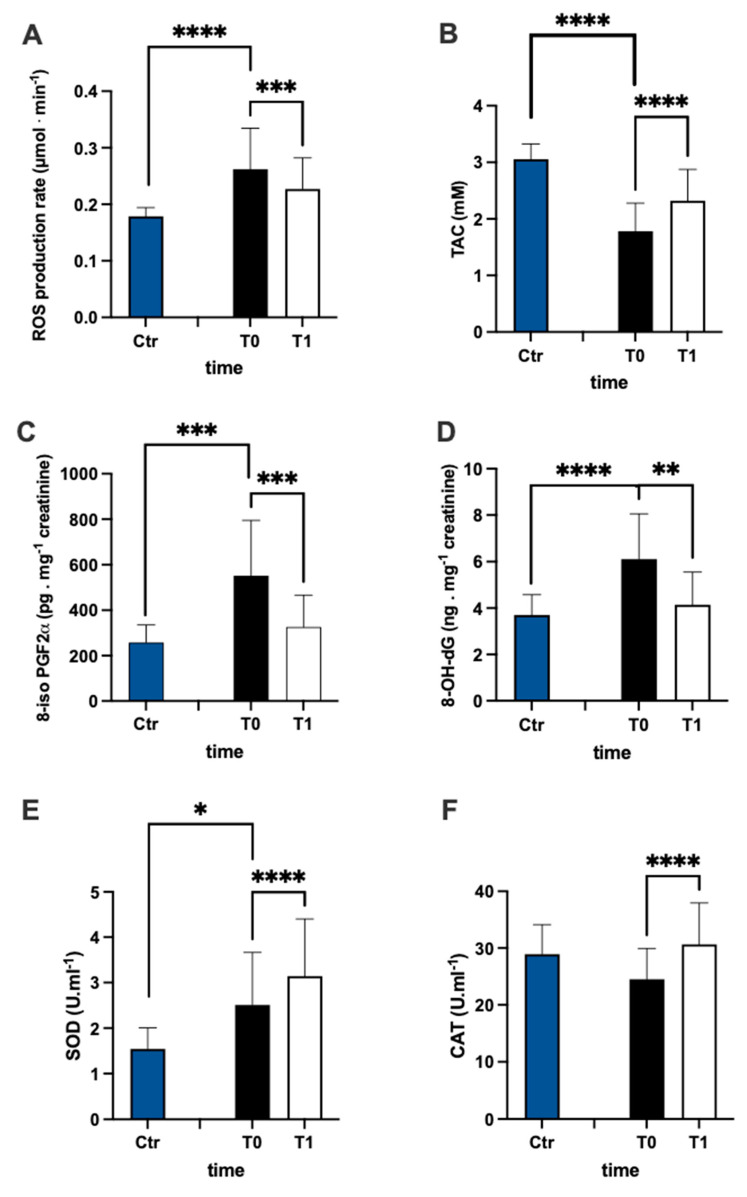
Bar charts (mean ± SD) in Ctr group and at T0, T1, in MS of (**A**) ROS production, (**B**) antioxidant capacity, (**C**) lipid peroxidation, (**D**) DNA oxidation, (**E**) superoxide dismutase, and (**F**) catalase. * *p* < 0.05, ** *p* < 0.01, *** *p* < 0.001, **** *p* < 0.0001, significantly different.

**Figure 3 antioxidants-12-01338-f003:**
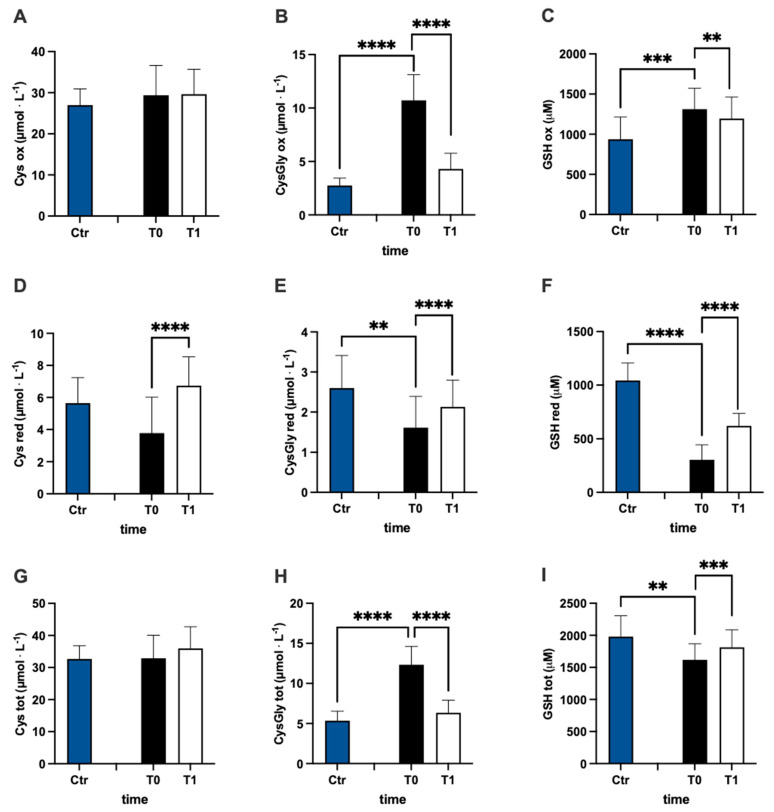
Bar charts (mean ± SD) in Ctr group and at T0, T1, in MS patients of cysteine (Cys) cysteinylglycine (CysGly) and glutathione (GSH) oxidized (**A**−**C**), reduced (**D**−**F**), and total (**G**−**I**), respectively, in RBC. Total homocysteine (Hcy) in plasma is reported too (**J**). ** *p* <0.01, *** *p* < 0.001, **** *p* < 0.0001, significantly different.

**Figure 4 antioxidants-12-01338-f004:**
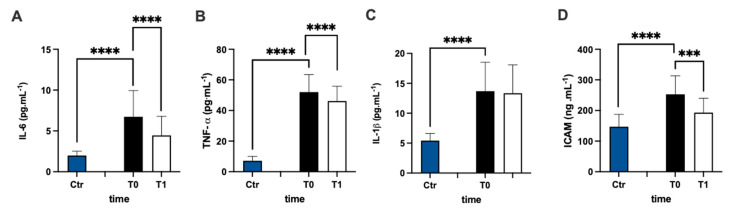
Bar charts (mean ± SD) in Ctr group and at T0, T1, in MS of (**A**) IL-6, (**B**) TNF-α, (**C**) IL-1β, and (**D**) intracellular adhesion molecules (sICAM). *** *p* < 0.001, **** *p* < 0.0001, significantly different.

**Figure 5 antioxidants-12-01338-f005:**
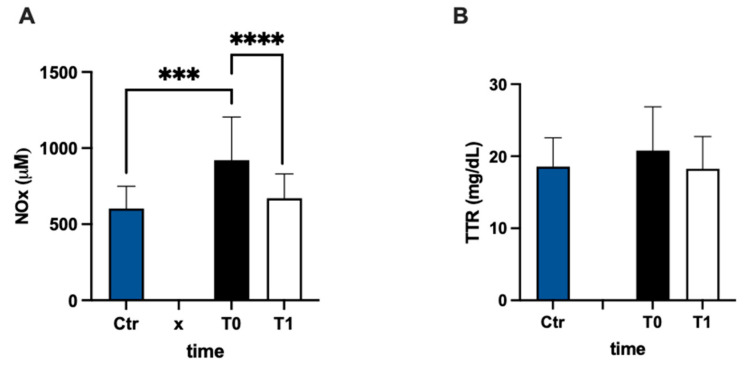
Bar charts (mean ± SD) in Ctr group and at T0, T1, in MS of (**A**) NO metabolites and (**B**) transthyretin. *** *p* < 0.001, **** *p* < 0.0001, significantly different.

## Data Availability

Data are contained within the article.

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
