# Peer review of "Chelation Therapy Associated with Antioxidant Supplementation Can Decrease Oxidative Stress and Inflammation in Multiple Sclerosis: Preliminary Results"

_antioxidants, 2023, doi:10.3390/antiox12071338_

Round 1

Reviewer 1 Report

The manuscript by Vezzeli et al. examines the effect of the combination of Chelation therapy associated with antioxidant supplementation in Multiple Sclerosis (MS) treatment. The authors observed that the treatment in MS induced a significant down-regulation of pro-inflammatory cytokines and biomarkers of lipid peroxidation. The protective effect may occur by decreasing ROS production and increasing antioxidant capacity.

The study presents interesting results but with very preliminary data. The number of individuals is low, and some of the statistical significances are difficult to believe, probably due to the tests used for multiple comparisons. In other cases, the graph does not accurately represent the values indicated in the text. For example, in figure 5A, it is not possible for the level of NOx in controls to be 57% lower than in MS.

I recommend revising the statistical analysis and increasing the number of individuals.

Author Response

The authors agree and thank the reviewer for his/her valuable observation. 

According to your suggestions, we revised all statistical data, and percent change.

Below the table reporting selected biomarkers analyzed in the study with p value and %.

In details:

delta % T control(=100%) vs d T0 MS

and

delta % T0 MS (=100%)vs T1 MS

 After normal distribution test, statistical analyses were performed using non-parametric test, in fact analysis shows a not Gaussian distribution. Non parametric test (no matching or pairing - multiple comparison - post hoc test: Dunn’s) was performed

To evaluate the effect of the chelation therapy and multivitamin treatment in MS subjects was used Wilcoxon matched-pairs signed rank test (T0 MS vs T1 MS), and d-Cohen (see new version manuscript)

Markers

Ctr

d%

(T0 vs Ctr)

ANOVA

p

(T0 vs Ctr)

T0

T1

d%

(T0 vs T1)

ANOVA

p

(T0 vs T1)

t-test

T0 vs T1

ROS

0,18

+44%

<0,0001

0,26

0,22

-15%

0,5715

0,0001

TAC

3,05

-42%

<0,0001

1,78

2,31

+30%

ns

<0,0001

8-iso

257,77

+114%

0,0010

552,13

326,25

-41%

ns

0,0005

DNA

3,70

+65%

0,0001

6,11

4,14

-32%

0,0182

0,0027

SOD

1,54

+63%

0,0111

2,51

3,14

+25%

0,0457

<0,0001

CAT

28,90

-15%

ns

24,52

30,66

+25%

0,0135

<0,0001

IL-6

1,99

+238%

<0,0001

6,73

4,45

-34%

ns

<0,0001

TNF-a

7,17

+626%

<0,0001

52,03

46,16

-11%

ns

<0,0001

IL-1b

5,43

+152%

<0,0001

13,69

13,34

-3%

ns

ns

ICAM

147,63

+71%

<0,0001

252,68

193,01

-24%

0.0463

0,0005

NOx

603,45

+53%

0.0010

921,07

670,18

-27%

0,0328

<0,0001

TTR

18,56

+12%

ns

20,8

18,27

-12%

ns

ns

GSH tot

1979,33

-13%

0.0016

1717,94

1814,52

+6%

ns

0.0003

CysGly tot

5,35

+130%

<0,0001

12,33

6,34

-49%

<0,0001

<0,0001

Cys tot

32,63

+1%

ns

32,88

35,99

+9%

ns

ns

Hcy tot

2,64

+185%

<0,0001

7,52

6,27

-17%

ns

ns

GSH ox

936,9

+40%

0,0007

1313,22

1195,64

-9%

ns

0,0079

CysGly ox

2,75

+290%

<0,0001

10,72

4,30

-60%

0,0005

<0,0001

Cys ox

26,99

+9%

ns

29,37

29,65

+1%

ns

ns

GSH red

1042,82

-71%

<0,0001

304,721

618,878

+103%

0,0186

<0,0001

CysGly red

2,59

-38%

0,0014

1,61

2,12

+32%

ns

<0,0001

Cys red

5,64

-33%

ns

3,78

6,73

+78%

0,0008

<0,0001

The text has been modified according these data.

We agree with reviewer’s observation that the study is preliminary and the number of subjects is low but the prospective calculations of power to determine sample size, made using G power software (GPower 3.1), made us confident in the data obtained. Moreover the aim of this preliminary study, after the confirmation of an altered oxy-inflammation status in MS patients examined, was to evaluate the possibility of an effect of the specific antioxidant treatment on oxy-inflammation status.

Thanks for the review and suggestions.

Attached new revised manuscript. Alessandra Vezzoli, and Co-authors  

Reviewer 2 Report

The authors described about the new therapy for multiple sclerosis using a chelation. It's very interesting. However, Imaging, especially MRI, is needed to assess whether or not multiple sclerosis responds to this therapy.

If this chelation therapy becomes a therapeutic drug of for multiple sclerosis, the changes before and after the treatment, such as lesional changes in MRI of the brain and spinal cord, may be necessary.

average

Author Response

Thanks for the review. We agree with reviewer’s observation but the aim of this preliminary study, after the confirmation of an altered oxy-inflammation status in MS patients examined, was to evaluate the effect of the specific antioxidant treatment on oxy-inflammation status. Indeed, an improvement of this latter is an advantage in every kind of disease condition. Considering the results obtained it is possible to plan MRI of the brain and spinal cord. A sentence about this possibility has been added in limitations paragraph.

Alessandra Vezzoli and co-Authors 

Round 2

Reviewer 1 Report

Without any additional comments, in my point of view, the manuscript could be accepted. However, I still believe that the statistical analysis is not the most appropriate